# Anticancer Effects of Thymoquinone through the Antioxidant Activity, Upregulation of Nrf2, and Downregulation of PD-L1 in Triple-Negative Breast Cancer Cells

**DOI:** 10.3390/nu14224787

**Published:** 2022-11-13

**Authors:** Getinet M. Adinew, Samia S. Messeha, Equar Taka, Ramesh B. Badisa, Karam F. A. Soliman

**Affiliations:** Division of Pharmaceutical Sciences, College of Pharmacy and Pharmaceutical Sciences, Institute of Public Health, Florida A&M University, Tallahassee, FL 32307, USA

**Keywords:** breast cancer, triple-negative breast cancer, oxidative stress, thymoquinone, Nrf2, PD-L1

## Abstract

The variety of therapies available for treating and preventing triple-negative breast cancer (TNBC) is constrained by the absence of progesterone receptors, estrogen receptors, and human epidermal growth factor receptor 2. Nrf2 (nuclear factor-erythroid 2-related factor), and PD-L1 (program cell death ligand 1), a downstream signaling target, have a strong correlation to oxidative stress and inflammation, major factors in the development and progression of TNBC. In this study, the genetically distinct MDA-MB-231 and MDA-MB-468 TNBC cells were treated with the natural component thymoquinone (TQ). The results show that TQ exhibits considerable antioxidant activity and decreases the generation of H_2_O_2_, at the same time increasing catalase (CAT) activity, superoxide dismutase (SOD) enzyme, and glutathione (GSH). Additionally, the results show that TQ treatment increased the levels of the different genes involved in the oxidative stress-antioxidant defense system PRNP, NQO1, and GCLM in both cell lines with significant large-fold change in MDA-MB-468 cells (+157.65 vs. +1.7, +48.87 vs. +2.63 and +4.78 vs. +2.17), respectively. Nrf2 mRNA and protein expression were also significantly increased in TQ-treated TNBC cells despite being higher in MDA-MB-468 cells (6.67 vs. 4.06). Meanwhile, TQ administration increased mRNA levels while decreasing PD-L1 protein expression in both cell lines. In conclusion, TQ modifies the expression of multiple oxidative-stress-antioxidant system genes, ROS, antioxidant enzymes, Nrf2, and PD-L1 protein, pointing to the therapeutic potential and chemopreventive utilization of TQ in TNBC.

## 1. Introduction

The most prevalent and often diagnosed cancer in women is breast cancer (BC). By the end of 2022, it is anticipated that there will be more than 287,850 new cases and 43,250 new fatalities in the US due to BC. Throughout their lives, the disease affects 13% (1 in 8) of all women [1]. An average of one woman is diagnosed with BC every two minutes, with the rate of increase being 0.5% per year [2]. Triple-negative breast cancer (TNBC) makes up 10–20% of all instances of BC [3]. It is frequently detected in advanced stages, is more likely to metastasize, and has a worse overall survival rate [4]. Progesterone receptor (PR), estrogen receptor (ER), and human epidermal growth factor receptor 2 (HER2) are not expressed in TNBC. The lack of these three receptors restricts the variety of therapies used to treat and prevent TNBC. Endocrine therapy and anti-HER2 (human epidermal growth factor receptor 2) drugs have dramatically increased the survival of patients with ER-positive and HER2-positive BC over the past ten years [5]. TNBC, however, still lacks potent treatment alternatives [6].

TNBC and other advanced aggressive breast cancers are strongly supported by the tumor microenvironment (TME) [7]. The TME has received significant research attention because of its ability to prevent immune surveillance and enable accelerated tumor formation and progression [8]. By secreting various inflammatory factors, growth factors, and matrix proteases or facilitating tumor angiogenesis and immunosuppression, they can either directly or indirectly promote the formation, growth, and spread of malignancies [9].

On the other hand, oxidative stress in TME increases ROS, DNA adduct production, proto-oncogene activation, and tumor suppressor gene suppression [10]. Cancer cells may produce more ROS because they lack the genes that typically have antioxidant defenses [11]. Breast cancer also supports their growth by promoting angiogenesis, creating hypoxia, and producing ROS [12]. Continuous production of ROS by the tumor cells causes increased mutation rates and accelerated tumor progression, activation of signaling pathways that promote growth, adaptation to oxidative stress leading to increased therapy resistance, increased blood supply to the tumor cells, and increased risk of metastasis [13]. Inflammatory mediators such as prostaglandin, NO, fatty acids, and ROS activate Nrf2 [14,15]. Although studies on the complex and varied nature of TNBC and the function of Nrf2 are now underway, there is not enough evidence to conclusively show that Nrf2 plays a substantial role in BC progression [16]. In BC cell lines and patient samples, Nrf2 is downregulated compared to healthy mammary epithelial cells [17]. Particularly in TNBC, Nrf2 was dramatically reduced in TNBC patients compared to non-TNBC patients, which may help explain the importance of the chemopreventive function of Nrf2 [18]. In addition, PD-L1 is one of the downstream targets of the Nrf2 signaling pathway, frequently expressed on tumor cell surfaces. TNBCs have higher quantities of PD-L1, which makes them a potential therapeutic target [19]. Positive PD-L1 expression in BCs is associated with large tumor size, higher tumor grade, a substantial number of lymph nodes, and negative ER and PR status [20,21]. In recent years, immunotherapy has been a promising treatment for TNBC.

Meanwhile, numerous investigations have discovered that natural and synthetic substances such as curcumin, xanthohumol, sulforaphane, and oltipraz activate Nrf2 expression, which prevents cancer [22,23,24,25]. Research has shown that Nrf2 activation decreases ROS and pro-inflammatory cytokines, reducing inflammation [14,26,27]. A protective effect against cellular damage is provided by Nrf2 signaling, which also raises the expression of genes involved in the anti-inflammatory and antioxidant response, increases the potential of the mitochondrial membrane, and improves mitochondrial function [28,29].

Many TNBC patients cannot be adequately treated or prevented with conventional therapy. Therefore, different strategies for treating TNBC are needed to develop novel therapeutic agents from bioactive natural compounds like TQ. TQ has been used to treat diseases, including cancer, for a long time [7]. The effects of TQ on various tumors have been thoroughly studied. In the human colon, breast, brain, pancreatic, ovarian, larynx, colon, myeloblastic leukemia, osteosarcoma, and lung cancer cell lines, TQ has been demonstrated to suppress cellular proliferation and induce apoptosis [3,30,31]. Additionally, it has been claimed that TQ produced cell cycle arrest, preventing BC from progressing from the G1 to the S phase by targeting the proteins cyclin E, cyclin D1, and p27 [32]. Along with protecting the activity of numerous antioxidant enzymes like catalase, glutathione peroxidase, and glutathione-S-transferase, TQ may also operate as a free radical and superoxide radical scavenger [33,34]. These results demonstrated that TQ’s mechanisms of action against different tumors are diverse.

The current study demonstrates the expression profiles of the genes associated with the antioxidant defense system against oxidative stress and the effects of TQ on Nrf2 and PD-L1 in MDA-MB-231 and MDA-MB-468 TNBC cells. Exploring the underlying mechanisms of anticancer actions of TQ as it relates to Nrf2 and PD-L1 associated with oxidative stress will provide more support for the possible use of TQ in the prevention/treatment of TNBC.

## 2. Materials and Methods

### 2.1. Materials

TQ was purchased from Sigma-Aldrich (purity 99%, cat # MKCC0600) (St. Louis, MO, USA). Trypsin-EDTA solution, penicillin/streptomycin, and phosphate-buffered saline (PBS) were purchased from the American Type Culture Collection (ATCC; Manassas, VA, USA). DPPH free Radical (cat # 1898-66-4) was purchased from Millipore (Burlington, MA, USA). From Millipore, DPPH free Radical (cat # 1898-66-4) was purchased (Burlington, MA, USA). We obtained DCFDA/H2DCFDA-Cellular ROS (cat # ab113851) from Abcam (152 Grove Street, Waltham, MA 02453, USA). We bought a DNA-free ^TM^ kit from Life Technologies, Inc. (Thermo Fisher Scientific, Inc., Waltham, MA, USA). The Human Oxidative Stress-Antioxidant defense PCR array H96, as well as primers specific for Nrf2, PD-L1, and GAPDH, were purchased from BioRad Laboratories. The iScript^TM^ cDNA Synthesis kit (cat. no. 170-8890), SsoAdvanced^TM^ Uni-versal SYBR^®^ Green Supermix, and the Human Oxidative Stress-Antioxidant defense PCR array H96 were also purchased (Hercules, CA, USA). Fetal bovine serum (FBS) and Dulbecco’s modified Eagle’s medium (DMEM) were procured from VWR International (Radnor, PA, USA). Opti-MEM was purchased from Life Technologies Corporation. GAPDH (14C10) Rabbit mAb HRP conjugate (cat# 3683s), PD-L1(E1L3N) XP Rabbit mAb (cat#13684s), and Nrf2 (E3J1V) Rabbit mAb (cat# 33649s) were used as primary antibodies in this study. The secondary antibody was Anti-Rabbit IgG, HRP-linked Antibody (cat#7074s). All antibodies were purchased from Cell Signaling Technology (3 Trask Ln, Danvers, MA 01923, USA). The primary antibody had the following molecular weights: GAPDH (37 kDa), Nrf2 (97–100 kDa), and PD-L1 (40–50 kDa).

### 2.2. Cell Culture

MDA-MB-231 (ATCC^®^ HTB-26TM) and MDA-MB-468 (ATCC^®^ HTB-132^TM^), two TNBC cell types, were acquired from ATCC and kept per ATCC’s maintenance instructions. Both cell lines were grown in 75-mL tissue culture (TC) flasks as monolayers at 37 °C in a humidified 5% CO_2_ incubator, occasionally subculturing with trypsin/EDTA. 4 mM L-glutamine, 10% heat-inactivated FBS (*v*/*v*), and 1% penicillin/streptomycin salt solution (100 U/mL and 0.1 mg/mL, respectively) were added to the complete growth DMEM. The experimental media was DMEM supplemented with 2.5% heat-inactivated FBS [35].

### 2.3. Assay for Scavenging DPPH Radicals

The 1,1-diphenyl-2-picrylhydrazyl (DPPH) radical-scavenging activity was used to test the antioxidative activity [36]. In brief, a 96-well plate was filled with 100 μL/well of freshly made DPPH methanolic solution (100 μM). To create a serial dilution of TQ (0–300 μM), a 10 mM stock solution of TQ was reconstituted in DMSO. The first well of a 96-well plate received 100 μL of a methanol-DMSO solution (1% DMSO), while the second well-received 10 μM of vitamin C as a positive control. Then, 100 μL of TQ was added to the remaining wells. A microplate reader (λ = 405 nm) was used to measure the absorbance following a 30 min incubation period at room temperature and complete darkness. The following formula was used to convert the DPPH data into a percentage of DPPH scavenging activity: % scavenging activity = [(A_blank_ − A _sample_)/(A_blank_)] ∗ 100.

### 2.4. Detection of Intracellular Reactive Oxygen Species Level (ROS)

The ability of TQ to lower intracellular levels of ROS in TNBC cells was examined using the fluorogenic dye 2′,7′-Dichlorofluorescin diacetate (DCFDA), as previously described [37]. To produce the highly fluorescent 2′ 7′-dichlorofluorescein, H2DCFDA was passively diffused into cells, where it was deacetylated by intracellular esterases into a nonfluorescent molecule (DCF). Prior to the experiment, cells were cultivated for 24 h in 96-well plates at a density of 2.5 × 10^4^ cells per well. After that, cells were exposed to TQ at 5, 10, and 15 µM for 24 h. The concentration was selected based on our previously published findings demonstrating that more than 80% of the cells were still alive after 24 h of TQ exposure [3]. The culture media was then replaced with 25 µM H2DCFDA in Opti-MEM, a reduced serum medium, and incubated for 45 min. A Synergy HTX Multi-Mode microplate reader (BioTek Instruments, Inc., Winooski, VT, USA) was used to measure the DCF fluorescence at 485 nm excitation and 520 nm emission [38], and a Nikon inverted microscope eclipse (Melville, NY 11747-3064, USA) was also used to measure the fluorescence image [39].

### 2.5. Hydrogen Peroxide Cell-Based Assay

The extracellular H_2_O_2_ produced by TNBC cells was measured using a hydrogen peroxide cell-based assay kit (Cayman Chemical, Arbor, MI, USA). Using horseradish peroxidase (HRP) as a catalyst, this extremely sensitive and stable H_2_O_2_ probe interacts with extracellular H_2_O_2_ to create the highly fluorescent resorufin. Briefly, cells were plated at a density of 5 × 10^4^ per 100 µL in 96-well plates in a serum-free culture medium. The cells were exposed to TQ at different concentrations (5, 10, and 15 µM) for 24 h after overnight incubation. For the treatment controls and catalase controls, extra wells for media and working solutions without cells were allotted, as instructed in the kit’s instruction booklet. According to the directions in the kit, the samples were examined. Using a microplate reader, the fluorescence intensity of each well (530 nm for excitation and 590 nm for emission) was measured.

### 2.6. Catalase Enzyme Activity Assay

In this study, we used the catalase enzyme activity assay kit, and we adhered to the instructions (Cayman Chemical). In brief, a serum-free medium was used to plate TNBC cells from the MDA-MB-231 and MDA-MB-468 strains in triplicates (5 × 10^5^ cells/well/6 well plates). Cells were exposed to TQ at different concentrations (5, 10, and 15 µM) over 24 h after overnight incubation. Cells were scraped, gathered, and centrifuged at 1000–2000× *g* for 10 min at 4 °C. The resulting cell pellets were sonicated in the proper ice-cold buffer (50 mM potassium phosphate, pH 7.0, containing 1 mM EDTA), centrifuged, and the supernatants were preserved for testing as per the instructions provided with the kit. A plate reader operating at 540 nm was used to measure the absorbance at the designated wavelengths.

### 2.7. Superoxide Dismutase Enzyme Activity Assay

The cellular antioxidant defense system depends on SOD, an antioxidant enzyme that catalyzes the dismutation of superoxide anion. Utilizing a tetrazolium salt from the Cayman Chemical Superoxide Dismutase Assay Kit (Cat#706002), total, cytosolic, and mitochondrial SOD activity was evaluated in order to identify superoxide radicals generated by xanthine oxidase and hypoxanthine. The simple method for this assay is the same as the set of the system in Section 2.6. The cell pellets were sonicated in the proper ice-cold buffer according to the kit’s guidelines, then centrifuged at the allocated speed and for the allotted period of time at 4 °C, with the supernatants being retained for assay in accordance with the instructions. The absorbance at the designated wavelengths was determined at 450 nm using a plate reader.

### 2.8. Glutathione Assay

The amount of total GSH in the sample was evaluated using an enzymatic DTNB (5, 5′-dithio-bis-2- (nitrobenzoic acid), Ellman’s reagent)-GSSG reductase recycling technique utilizing the Cayman Chemical Glutathione assay kit (Cat# 703002). By using this method, a 5-thio-2nitrobenzoic acid is produced, which is a yellow substance that is directly proportional to the total amount of GSH. In summary, as in Section 2.6, cells from 6-well plates that had been cultured and treated were removed using a cell scraper and centrifuged at 1000–2000× *g* for 10 min at 4 °C. In 1 to 2 mL of cold, 50 mM MES buffer (pH 6–7), 1 mM EDTA-containing solution, the pellet was sonicated into a homogenous supernatant before being centrifuged at 10,000× *g* for 15 min at 4 °C. The supernatants were stored on ice with an equal amount of freshly prepared, 10% metaphosphoric acid (MPA) reagent for deproteinization. To prepare the samples for measurement, freshly prepared 4 M TEAM reagent was added to them. The experiment was conducted in accordance with the kit’s instructions, and an absorbance between 405 and 414 nm was measured.

### 2.9. Gene Expression Assay

#### 2.9.1. RNA Extraction and cDNA Synthesis

Using previously described procedures, profiling the expression of the major antioxidant genes was developed in MDA-MB-231 and MDA-MB-468 TNBC cells [40]. In brief, TQ was applied for 24 h after each cell line had been cultured for an overnight period at 37 °C (at a density of 10 × 10^6^ cells/mL in T-75 flasks). Cells were mechanically removed from each flask following a 24 h treatment, pelleted, and given two PBS washes. Total RNA was extracted from each cell pellet using 1 mL of the TRIzol reagent, as directed by the manufacturer. For phase separation, 0.2 mL of chloroform was added to each sample, vortexed, and centrifuged for 15 min. at 10,000× *g* and 2–8 °C after incubating at ambient temperature for 2–3 min. The aqueous phase was collected and combined with 0.5 mL of isopropyl alcohol to pellet the RNA. Following a 70% ethanol wash, the RNA pellets were reconstituted in 20 μL of nuclease-free water and stored at −80 °C for further use. A NanoDrop spectrophotometer was used to measure the amount and quality of RNA (NanoDrop Technologies; Thermo Fisher Scientific, Inc., Waltham, MA, USA). The iScriptTM cDNA Synthesis kit was then used to create the cDNA for each sample, which was then kept in a freezer at −80 °C.

#### 2.9.2. Quantitative Real-Time PCR Arrays

The SsoAdvanced^TM^ Universal SYBR^®^ Green Supermix and reconstituted cDNA (2.3 ng) were added into each well of the 96-well human oxidative stress-antioxidant defense system array, and the plate was shaken for 5 min before being centrifuged at 1000× *g* for 1 min. The Bio-Rad FX96 Real-Time System (Bio-Rad Laboratories, Hercules, CA, USA) was used to set up the quantitative fluorescence PCR run with 39 thermo-cycling denaturations: 30-s activation at 95 °C, 10-s denaturation at 95 °C, 20-s annealing at 60 °C, and 31-s extension at 65 °C [3]. The RT-PCR outcomes for each cell line were validated by three separate experiments. The Student’s *t*-test with GraphPad Prism (v 9.3.1) software was used to further corroborate the results after the gene expression was assessed using the CFX 3.1 Manager software (Bio-Rad Laboratories, Hercules, CA, USA) program.

#### 2.9.3. Quantitative RT-PCR

To examine mRNA gene expressions, quantitative Real-Time PCR tests for PD-L1 and Nrf2 mRNA were run. Consistently, cells were exposed to various TQ concentrations, and the cDNA was produced as described previously (Section 2.9.1). With the same PCR run, the expression levels of the genes of interest were determined (Section 2.9.2). Gene expression levels were scaled back to GAPDH. All reactions were conducted in triplicate.

### 2.10. Capillary Electrophoresis Western Analysis

As previously described, a protein quantification assay was conducted using capillary-based Western blot analysis [41,42]. The cells were stimulated with INF-Y (100 nM), collected, and centrifuged after being cultured with both cell lines at a density of 1 × 10^7^ cells/T-75 flasks for the specified time. Each sample’s acquired pellet was rinsed with ice-cold PBS before being resuspended in protease inhibitor cocktail-infused RIPA buffer (Thermo Fisher, Waltham, MA, USA) at 4 °C for 10 min. The bicinchoninic acid (BCA) protein assay kit measured the protein concentration (Thermo Fisher, Waltham, MA, USA). Samples containing 0.6 mg/mL of protein for Nrf2 and 1 mg/mL for PD-L1 were used to calculate after an initial optimization assay. The dilution factor for the two antibodies under examination was 1:125 for the MDA-MB-231 and MDA-MB-468 samples. GAPDH’s dilution factor, meanwhile, was 1:50. The microplate was loaded and put in the device in accordance with the manufacturer’s instructions (Protein Simple, Bio-Techne, 614 McKinley Place NE, Minneapolis, MN 55413, USA). The reaction took place inside the capillary system and utilizing the aforementioned antibodies; it was possible to determine the chemiluminescence response, identify the protein expression, and capture digital blot photographs. *GAPDH* was used as an internal control.

### 2.11. Statistical Analysis

GraphPad Prism 9.3.1 was used to analyze the data for this study (GraphPad Software, Inc., San Diego, CA, USA). The average of at least three separate experiments is represented by the mean ± S.E.M. of all data points. As shown in the legends, a one-way analysis of variance (ANOVA) was used to establish the significance of the difference, and then Bonferroni’s multiple comparison test. The unpaired Student’s t-test was applied to compare two data sets. At *p* < 0.05, a difference was deemed significant (as indicated in the figures and legends).

## 3. Results

### 3.1. Antioxidative Effect of Thymoquinone by Scavenging DPPH

In vitro, an antioxidant assay of TQ revealed the presence of antioxidant potential. TQ’s radical scavenging action was validated using the DPPH assay. TQ converted the stable radical DPPH to the yellow diphenyl picrylhydrazine. As shown in Figure 1, a low concentration of TQ (<4.6 µM) did not induce a significant effect on DPPH scavenger activity. Meanwhile, a concentration-dependent significant impact (*p* < 0.05–0.0001) was detected at higher concentrations, reaching 27.2% of DPPH scavenging activity at 300 µM of TQ. Adding vitamin C as a positive control resulted in a 28.22% reduction in DPPH at 10 µM.

### 3.2. Effect of TQ on the Intracellular ROS Level of TNBC Cells

The DCFDA assay was then used to assess the intracellular production of ROS in TNBC cells. Our study revealed that in the MDA-MB-231 TNBC cell line (Figure 2A), intracellular ROS generation was reduced by 10 (*p* = 0.0321), 15 (*p* = 0.0061), and 27% (*p* = 0.0004) at concentrations of 5, 10, and 15 µM, respectively, whereas in MDA-MB-468 cells, they were reduced by 8 (*p* = 0.0015), 20 (*p* < 0.0001), and 29% (*p* < 0.0001) at the same concentrations (Figure 2B).

### 3.3. Thymoquinone Decreases Hydrogen Peroxide Levels in TNBC

Following TQ treatment, hydrogen peroxide levels in MDA-MB-231 and MDA-MB-468 TNBC cells indicated a significantly different response. Compared with the DMSO-treated control cells, the hydrogen peroxide levels in the TQ-treated MDA-MB-231 TNBC cells have not shown any significant difference (Figure 3A). However, in MDA-MB-468 TNBC cells, TQ at 5 and 10 µM induced a significant decrease in H_2_O_2_ concentrations (*p* < 0.0001) compared to control (53.5 vs. 32.0 and 33.8, respectively), meanwhile a nonsignificant effect was observed at the highest tested concentrations (15 µM) (Figure 3B).

### 3.4. Thymoquinone Increased Catalase Enzyme Activities in TNBC Cells

TQ significantly increased CAT activity in MDA-MB-231 TNBC cells (Figure 4A). However, there was no discernible change between the TQ-treated cells in MDA-MB-468 TNBC cells compared to the control group (Figure 4B). In MDA-MB-231 cells, TQ increased catalase activity in a concentration-dependent manner, as shown in Figure 4A, compared to the DMSO-treated group with a value of 0.13 vs. 0.19 vs. 0.20 and 0.22 nmol/min/mL at 0, 5 (*p* = 0.03), 10 (*p* = 0.029) and 15 (*p* = 0.016) µM, respectively. However, in MDA-MB-468 TNBC cells, TQ does not show any significant difference compared to the control for catalase activity.

### 3.5. Increased Superoxide Dismutase (SOD) Enzyme Activities in Thymoquinone-Treated TNBC Cells

MDA-MB-231 cells-treated TQ showed increased SOD activity levels in the cytosol and mitochondria (Figure 5A). In contrast, SOD activity in the mitochondria was not changed in the TQ-treated MDA-MB-468 cells compared with DMSO-treated control cells. Meanwhile, SOD activity only increased in the cytosolic at the highest tested concentration (15 µM) (Figure 5B).

### 3.6. Thymoquinone Increases Levels of the Antioxidant Glutathione

Compared to the control group, a significant induction of the total GSH and GSSG levels was measured in TQ-treated MDA-MB-231 cells (Figure 6A). A gradual and significant increase in GSH (*p* < 0.01 and 0.0001) and GSSG (*p* < 0.0001) levels were measured by increasing the tested concentrations, reaching a five and two-fold upregulation in GSH and GSSG, respectively, at 10 µM. This effect was decreased at the highest used concentration (15 µM). However, still higher than the control (*p* < 0.01 and 0.0001, respectively). Compared to the control group, the GSH level in TQ-treated MDA-MB-231 TNBC cells significantly induced expression of total GSH and reduced GSSG (Figure 6A). In contrast, a mixed effect was shown in TQ-treated MDA-MB-468 cells. As presented in Figure 6B, an inverse relationship was found between GSH levels and the tested concentrations. Indeed, the treated cells exhibited a 5-fold to 10-fold reduction in GSH level (*p* < 0.0001). Interestingly, a significant increase in GSSG was only found at 5 µM (*p* < 0.01), followed by a 50% significant reduction (*p* > 0.001) in GSSG at 15 µM of TQ.

### 3.7. Oxidative Stress-Antioxidant Defense Gene Expression Alteration in TQ-Treated Triple-Negative Breast Cancer Cells

TQ-treated TNBC cells were subjected to quantitative real-time PCR to determine the transcriptome level of oxidative stress-antioxidant-related genes to investigate the cytoprotective mechanism of the compound. Both cell lines were consistently treated with 15 µM of TQ. Profiling normalized mRNA expression for the cells under study revealed TQ’s impact on numerous oxidant-antioxidant-linked genes; however, we only displayed the significantly altered mRNAs. The red dots represent upregulated genes, the green dots represent downregulated genes, and the black dots in both cell lines indicate that gene expression is unchanged (Figure 7A,B). Additionally, compared with MDA-MB-231 cells, fewer genes were significantly altered in MDA-MB-468 cells. Notably, this alteration in the mRNA expression was exhibited with an outstandingly higher fold (+/−). Treating MDA-MB-231 cells for 24 h with 15 µM of TQ induced a significant downregulation in the mRNA of 11 genes, accompanied by a significant upregulation in the expression of 5 other genes (Figure 7A). In its counterpart, MDA-MB-468 cells, four genes were significantly upregulated, and three genes were repressed (Figure 7B).

Given those three genes; *NQO1, GCLM*, and *PRNP, were* upregulated in both cell lines, a tremendous increase in *PRNP* (+157.65-fold) was found in MDA_MB-468 cells compared with MDA-MB-231 cells (+1.70-fold). Likewise, in NQO1 (48.87 vs. 2.63-fold) and GCLM (4.78 vs. 2.17). On the contrary, *GPX1* was upregulated by 20.87-fold in MDA-MB-468 but downregulated in MDA-MB-231 cells (−2.54-fold). Furthermore, the mRNAs of three more genes were repressed in MDA-MB-468 cells, whereas *SEPP1* was the most inhibited (−15.84-fold). Indeed, elven genes were significantly inhibited in MDA-MB-231 cells, but the fold change did not exceed 4-fold. Additionally, less than threefold upregulation was measured in five more genes (Table 1). This study suggests that TQ has significantly impacted the expression of various oxidative stress-antioxidant defense system genes on MDA-MB-231 and MDA-MB-468 TQ TNBC cells compared to non-treated groups. Table 1A, B show the fold-change of the genes in TQ-treated TNBC cells directly generated by the Bio-Rad software.

When comparing the expression levels of the most significantly altered genes in MDA-MB-231and MDA-MB-468 TNBC cells, following the student t-test analysis compared to control, TQ significantly reduces the mRNA expression levels of *ATOX1*, *PNKP*, and *NUDT1* by more than 2-, 3.7-, and 1.8-fold, respectively among the downregulated genes in MDA-MB-231 (Figure 8A–K). TQ, on the other hand, increased the mRNA expression levels of *GCLM*, *SRXN*, *TXNRD1*, *PRNP*, and *NQO1* by more than 2-, 1.5-, 1.5-, 1.5-, and 2-fold, respectively, in MDA-MB-231 (Figure 8L–P). In MDA-MB-468 cells, TQ significantly increased mRNA expression *GCLM*, *GPX1*, *PRNP*, and *NQO1* by more than 4.6-, 17.5-, 131-, and 37.7-folds, respectively (Figure 8Q–T). While TQ substantially reduced the gene expression of *SEPP1*, *SIRT2*, and *NOX5* by more than 8.8-, 11.7-, and 5-folds, respectively (Figure 8U–W). This study suggests that TQ has shown a significant impact on the expression of various oxidative stress-antioxidant defense system genes on MDA-MB-231 and MDA-MB-468 TQ TNBC cells compared to non-treated groups.

### 3.8. TQ Increases mRNA Expression of Nrf2 in TNBC Cells

We further investigated the mRNA expression of *Nrf*2 in both cell lines. Using the TCGA dataset, we first examined the expression of Nrf2 in BC samples. Using the UALCAN website tool, we discovered that the mRNA expression level of *Nrf2* was significantly lower in TNBC patients than in normal breast tissue. Furthermore, it was considerably lower in AA compared to CA (Figure 9A,B). We then examined *Nrf2* mRNA levels in MDA-MB-231 and MDA-MB-468 TNBC cells to confirm this finding. Our non-treated group results were consistent with the TCGA dataset, and TQ significantly increased the expression of Nrf2 (Figure 9C,F). The expression of *Nrf*2 was increased by 4.8-fold in MDA-MB-231 TNBC cells (Figure 9C), while it was increased by 11.4-fold in MDA-MB-468 cells (Figure 9F) at 15 μM of TQ and showed a concentration-dependent manner increase of *Nrf*2 expression. Even though *Nrf*2 expression was high in both cells compared to controls, MDA-MB-468 had the greatest fold change of *Nrf*2, 11.4 vs. 4.8, a more than 6-fold increase. 

### 3.9. TQ Increases the Expression of PD-L1 mRNA

We also investigated the levels of *PD-L1* mRNA in both cell lines. *PD-L1* mRNA expression was increased in both cell lines. In MDA-MB-231 cells, *PD-L1* expression increased by 1.75-fold (Figure 10 A–C), while in MDA-MB-468 cells, it increased by 3.67-fold (Figure 10 D–F).

### 3.10. TQ Induces Nrf2 Protein Expression in MDA-MB-231 and MDA-MB-468 TNBC Cells

Next, we investigate Nrf2 protein expression in TNBC cells, which is both directly and indirectly associated with the production of ROS, antioxidant enzymes, and genes that protect against oxidative stress. TQ’s stimulatory effect on Nrf2 protein expression after a 24 h administration was assessed using Western analysis. Compared to the control (*p* = 0.0003), Nrf2 was found to be four times higher (4.06-fold change) in MDA-MB-231 cells (Figure 11A,B, Appendix A). According to the findings, INF-γ slightly decreased the expression of Nrf2 in MDA-MB-231 and MDA-MB-468 cells. Cells treated with INF-γ and TQ had a three-fold (3.31) increase in Nrf2 expression (*p* = 0.0025) as compared to cells treated with the stimulant alone. INF-γ reduced Nrf2 expression was five times higher in the MDA-MB-231 cell line than in the MDA-MB-468 cell line, despite Nrf2 expression being higher in the MDA-MB-468 cells (Figure 11C,D Appendix A). TQ co-treatment boosted Nrf2 expression in both MDA-MB-231 and MDA-MB-468 cells, with a more significant increase in MDA-MB-468 cells. TQ raised the expression of *Nrf*2 in MDA-MB-468 cells by more than sixfold (6.67 percent) compared to the control (*p* = 0.0019). When the stimulant was co-treated with TQ, it rose 6-fold compared to the stimulant alone (*p* = 0.0005) (6.23). MDA-MB-468 had higher TQ Nrf2 expression than MDA-MB-231, with 6.67-fold and 4.06-fold increases in expression, respectively, compared to the control. The high expression of Nrf2 in MDA-MB-468 compared to MDA-MB-231 was consistent with the mRNA expression results from the PCR analysis. The data were normalized with *GAPDH* as an internal control.

### 3.11. TQ Inhibits PD-L1 Protein Expression in MDA-MB-231 and MDA-MB-468 TNBC Cells

In contrast to the mRNA expression, protein expression of PD-L1 was reduced in MDA-MB-231 cells after treatment with TQ (Figure 12A,B, Appendix A). When compared to the control (*p* = 0.048), it was significantly reduced by more than 2-fold (2.09). TQ reduced PD-L1 expression by 61 percent compared to the control (*p* = 0.049) in MDA-MB-468 (Figure 12C,D, Appendix A). PD-L1 expression was reduced by more than two-fold (*p* = 0.049) relative to the control. In contrast to the mRNA expression, protein expression results of *PD-L1* in both cell lines were decreased. This effect could imply that TQ regulates PD-L1 protein post-translational modification rather than mRNA. The date was normalized with *GAPDH* as an internal control.

## 4. Discussion

The tumor microenvironment (TME) is a fundamental mediator for tumor cells to communicate with nearby cells through the lymphatic and circulatory systems. It significantly contributes to promoting TNBC, an advanced aggressive breast cancer. Clinical prognosis is poor for patients with solid tumors with significant TME, such as breast, prostate, cervical, and ovarian cancer [43]. Oxidative stress and immune system perturbations correlated to Nrf2 and its downstream signaling pathway, PD-L1, are metabolic features of the tumor microenvironment [14,44,45].

Numerous human diseases, including cancer, are influenced by oxidative stress, and higher ROS levels have been found in many malignancies [46] [47]. ROS plays a role in tumor stemness maintenance, angiogenesis, energy consumption, cell motility, and the progression and proliferation of the cancer cell cycle [48,49]. Additionally, ROS stimulates signaling pathways involved in the spread of tumors [50]. The continual production of free radicals by cancer cells results in cell damage because the antioxidant defense systems of cells have been impaired. All biological systems include defensive mechanisms against damaged molecules, such as antioxidants; however, they are not always successful [51]. According to the current investigation results, TQ possesses potent antioxidant activity in the DPPH and DCFH-DA assays. Data from the DPPH assay revealed that TQ had a significant antioxidant capacity and may be a potent free radical scavenger. The ability of TQ to scavenge DPPH radicals was concentration-dependent, with TQ showing considerable antioxidant effects at the highest concentration examined in this study (300 µM), comparable to those of the common positive control ascorbic acid. Furthermore, the TQ decreased TNBC’s endogenous ROS production results in this study are consistent with earlier research. Previous studies have shown the whole Nigella Sativa extract’s antioxidant activity and radical-scavenging capacity [52]. Similarly, several investigations suggested TQ as a potent superoxide anion scavenger [53,54]. TQ has also been demonstrated to protect against tissue damage caused by free radicals in earlier in vivo experiments [55]. The same study also found that TQ was more effective as a superoxide anion scavenger than the synthetic compound TBHQ (tert-butylhydroquinone), which has a similar chemical makeup [56]. According to these results, TQ is a radical scavenger that may help treat TNBC patients and avoid oxidative stress. TQ might therefore be a great source of natural antioxidants and a potential supplement for shielding cells from the free radical damage brought on by TNBC.

On the other hand, H_2_O_2_ is a strong oxidant but is not very reactive because of the delayed kinetic interactions with many biomolecules [57]. In the current study, H_2_O_2_ levels in the control group were not overly high in the MDA-MB-231 cells, and TQ successfully kept levels normal. The level of H_2_O_2_ in the control group was higher than usual in MDA-MB-468 cells. H_2_O_2_ expression was significantly reduced by TQ, notably at 5 and 10 μM. Inside and outside cells, H_2_O_2_ functions as a biological signaling molecule that increases the production of adhesion molecules, regulates cell proliferation or apoptosis and modifies platelet aggregation [58]. Therefore, eliminating excess H_2_O_2_ by antioxidant enzymes is crucial in preventing cellular damage. One of the primary enzymes that remove H_2_O_2_ is catalase. Although catalase activity varies greatly among cancer cell lines, cancer cells often have low CAT levels [59]. Compared to the control, TQ in the current study increased catalase activity in MDA-MB-231 cells in a concentration-dependent manner but had no discernible effect on MDA-MB-468 TNBC cells. On the other hand, the CAT, SOD, and GPX activity enzymes dramatically decreased in 20, 40, and 60 μM TQ compared to control cells, according to earlier research on the A549 lung cancer cell line [60].

Meanwhile, TQ enhanced SOD activity in MDA-MB-231′mitochondria at 15 μM as well. In both cell models, the influence of glutathione produced a different response. TQ increased total and oxidized glutathione levels in MDA-MB-231 cells at specific doses, but MDA-MB-468 cells had lower levels. Previous studies have shown that glutathione metabolism can affect cancer in beneficial and detrimental ways [61,62]. A change in this pathway can majorly impact cell survival and is crucial for detoxifying and removing toxins. By making tumor cells resistant to some chemotherapeutic drugs, increased glutathione levels can protect them [63]. Previous investigations in our laboratory have shown that TQ significantly reduced H_2_O_2_ production in BV2 microglial cells while also considerably lowering glutathione levels [64]. These findings suggest that TQ may affect antioxidant enzyme control in diverse cancer subtypes differently. According to our study, genetic variations may have contributed to the variable regulation of the antioxidant enzymes in TQ-treated TNBC cells.

The gene expression analysis of the oxidative stress-antioxidant defense system revealed that TQ dramatically decreased 11 and elevated 5 genes in MDA-MB-231 TNBC cells. TQ affects 3 and 4 genes differently in MDA-MB-468 TNBC cells, up- and down-regulating them. In both cell lines, the expression of three genes—NQO1, PRNP, and GCLM—was increased. We uncovered the considerably changed genes in both cell lines, their role in the emergence of different malignancies, including BC, and how they may be employed as a possible target in treating TNBC. In the current investigation, the data show that TQ treatment dramatically boosted the expression of NADPH: Quinone oxidoreductase 1 (NQO1), which was previously observed to be low in MDA-MB-231 and MDA-MB-468 TNBC cells. NQO1 is a flavoenzyme expressed in various tissues and can be found in the cytosol [65]. Both normal and oxidative stress situations are under the control of the antioxidant response element (ARE) [51]. The NQO1 gene, which has ARE in its promoter region, is regulated by Nrf2 [66]. Along with other Nrf2-induced detoxifying enzyme genes like GST (glutathione S-transferase) and HO-1 (heme oxygenase), it has been demonstrated that the NQO1 gene is activated in response to oxidants, ionizing radiation, xenobiotics, heat shock, electrophiles, hypoxia, and heavy metals [65,67]. Due to its ability to prohibit quinones from going through one-electron reduction, which would otherwise produce semiquinone free radicals and ROS, NQO1 is an anticancer enzyme [68]. To prevent cancer, nutritional supplements that stimulate NQO1 expression have gained popularity [65]. Based on our current findings, TQ targeting *NQO1* could be a potential therapeutic option in treating and preventing TNBC.

Additionally, we found that TQ boosted the expression of Prion protein (PRNP) in MDA-MB-231 and MDA-MB-468 TNBC cells. According to earlier research, PRNP function as a free radical scavenger and/or oxidative stress sensor molecule, according to several studies that corroborate our current findings [69]. In vitro tests on prostate tumor spheroids revealed a relationship between ROS levels and elevated PRNP expression. These findings imply that intracellular redox status and PRNP expression in tumors are related and may enhance antioxidative defense [70]. PRNP elevation may happen due to stress to increase cellular antioxidant capacity and engage in oxidative insult competition. To sum up, this study raises the possibility that PRNP may be involved in the cellular antioxidative system. Further investigation is required to ascertain the functions of PRNP and its isoforms in cancer.

TQ treatment boosted GCLM expression in MDA-MB-231 and MDA-MB-468 TNBC cells. Nrf2-mediated modulation may explain TQ-induced GCLM expression. Previous studies have demonstrated that tumor cell GCLM overexpression causes cellular resistance to oxidative stress [71]. In line with this finding, glutathione levels in cultured pancreatic mouse islet cells drop when GCLM mRNA is downregulated by ribozyme expression [72]. The transcription factor Nrf2 specifically targets the antioxidant response element sequence, which is present in GCLM promoters [73]. According to this study, GCLM may help the cellular antioxidant system prevent and treat TNBC.

Moreover, TQ boosted GPX1 in MDA-MB-468 cells while decreasing GPX1 in MDA-MB-231 TNBC cells, suggesting that TQ may control GPX1 differentially in genetically distinct TNBC cells. A phylogenetic family of enzymes known as the GPXs system functions in several crucial biological settings [74]. GPX1 has been associated with carcinogenesis and the progression of cancer. In some malignant tumors, the increased expression of GPX1 is associated with carcinogenic outcomes [75]. The ability of GPX1 to lessen oxidative DNA alterations has led to the intriguing finding that it may have a protective function in the early stages of carcinogenesis [76]. Additionally, GPX4 has been suggested as a biomarker of elevated BC risk [76,77,78,79] and is crucial for regulating apoptosis [74]. Again, *GPX4* has been found to be significantly associated with BC survival in patients with Native American ancestry. Although further studies are needed, we propose that *GPX1* and *GPX4* could be used as potential targets in TNBC cells.

Many studies have shown that Nrf2 activation decreases inflammation by lowering ROS and producing pro-inflammatory cytokines [14,80,81]. A protective effect against cellular damage is provided by Nrf2 signaling, which also raises the expression of genes involved in the anti-inflammatory and antioxidant response, increases the potential of the mitochondrial membrane, and improves mitochondrial function [82,83]. The Nrf2 gene is crucial for controlling ROS brought on by oxidative stress-related illnesses like cancer. Nrf2 is an essential transcriptional master regulator to maintain redox equilibrium that encourages the synthesis of antioxidant and cytoprotective genes in cells [84]. Numerous detoxifying and antioxidant genes are regulated by Nrf2 both at their basal and induced levels [85]. Cancer is one of many diseases that Nrf2 helps to avoid by regulating the basic cellular defense mechanisms [86]. Numerous research shows that Nrf2 is a good candidate for cancer prevention. Increased intracellular ROS levels in untreated groups demonstrated that TNBC increases oxidative stress by suppressing Nrf2 activation. In both cell lines, TQ dramatically increased the expression of Nrf2 on the mRNA and protein levels, according to our study. According to statistics from the UALCAN website, Nrf2 expression is decreased in TNBC compared to non-TNBC patients [18]. Compared to the non-treated group, the cell with 15 µM of TQ had the highest amount of Nrf2 mRNA and protein expression. TQ therapy reversed the Nrf2 suppression caused by TNBC. The expression of Nrf2 was upregulated at a rate of 6.67 and 4.06 folds, respectively, higher in MDA-MB-468 than in MDA-MB-231. TQ will be more helpful to MDA-MB-468 patients, according to the UALCAN web-based data, which revealed that Nrf2 was downregulated more in AA women than in CA women [18]. One of the key routes for reducing oxidative stress has been identified: the Nrf2 antioxidant-responsive element (ARE) pathway [87]. TQ dramatically boosted the amounts of Nrf2, one of the specific proteins in the Nrf2/ARE signaling pathway, by over- and under-expressing genes involved in oxidative stress and antioxidant defense. To maintain redox equilibrium, Nrf2 regulates several pathways for synthesizing antioxidants [29]. Nrf2 controls the expression of antioxidant and detoxifying genes in healthy cells in response to xenobiotic and oxidative stress [73].

Immune surveillance, including innate and adaptive immune responses, is crucial to cancer development [88]. Through tumor immune escape, tumor cells acquire strategies for eluding host defenses in the tumor microenvironment [89,90]. An essential component of the tumor immune escape mechanism is the PD-1/PD-L1-mediated immunological checkpoint in the TME [91]. Malignancies may avoid the antigen-specific T-cell immune response by activating the PD-1/PD-L1 signaling pathway [92,93]. To stop immune cells from destroying tumor cells, PD1 on immune cells interacts with PD-L1 on tumor cells [94,95]. TQ has been found to alter the expression of PD-L1 in TNBC cells. Increased or decreased ROS generation had no simple, direct relationship to the TME’s modulation of PD-L1 expression. PD-L1 expression tends to rise with increased ROS generation [96]. It is hypothesized that TQ scavenging of ROS reduces PD-L1 expression through Nrf2 activation. A recent study revealed that MDA-MB-231 and MDA-MB-468 TNBC had significantly higher levels of PD-L1 expression. Here, we demonstrated that TQ boosts mRNA expression in both cell lines while decreasing PD-L1 protein expression. TQ may impact post-transcriptional processes, such as mRNA stability, translation, or protein degradation, according to the decrease in PD-L1 protein expression. Studies have shown that post-transcriptional processes control protein levels without regard to the quantity of mRNA [97].

Additionally, TQ’s effects on the following processes could account for the negative correlation: protein transport, protein synthesis delay, protein half-life regulation, and translation rates [98]. More research is needed to confirm whether TQ influences these specific pathways between *PD-L*1 mRNA and protein levels. This is not a novel concept in cancer treatment. Some drugs act as translational-specific inhibitors in cancer treatment, such as rapamycin, an mTOR inhibitor, one of the most critical translational regulators that impair cancer metabolism and has been extensively researched as a cancer drug [99].

## 5. Summary

Oxidative stress and inflammation in the TME aggravate several different cancers. Numerous malignancies, including TNBC, have high ROS levels and inflammatory mediators. Due to weakened cellular defense mechanisms, the continual production of free radicals and pro-inflammatory chemicals by cancer cells in the TME led to cell damage. TQ exhibits a significant level of antioxidant activity in in vitro tests. Both TQ-treated cell types showed enhanced Nrf2 mRNA and protein expression. The modulation of many oxidative stress-antioxidant defense enzymes, which are altered either directly or indirectly by Nrf2, is a mechanism through which TQ exerts its cytoprotective effects. TQ may be a valuable target in treating TNBC due to its capacity to upregulate and downregulate several biomarkers signaling pathways. Few Nrf2 activators are utilized in clinics, although they have the potential to be used as possible treatments for a variety of diseases [100]. Therefore, it is recommended to use safe Nrf2 activators for clinical application. As a result, it is possible to suggest TQ as a novel Nrf2 activator for application in breast cancer therapy. Finally, the PD-L1 immunological checkpoint can be significantly impacted by TQ-induced ROS levels. The current study provides the molecular basis for TQ utilization in developing novel therapeutic combinations that include anti-PD-L1 and particular ROS modulators like Nrf2. The total oxidative-stress-antioxidant defense system genes controlled by TQ in MDA-MB-231 and MDA-MB-468 TNBC cells are summarized in Figure 13.

## 6. Conclusions

TQ-mediated cytoprotection involves the activation of *Nrf*2, down-regulation of PD-L1, and the subsequent modulation of several cytoprotective enzymes. These sequential events suggest a mechanism in which *Nrf*2, *PD-L1*, and cytoprotective enzymes work together to protect cells from oxidative stress and inflammation involved in the TME in the presence of TQ. Many aspects of this TQ-induced cellular modulating pathways require further investigation. Because of the aggressive nature of the disease and the lack of targeted therapies, our findings regarding the oxidative stress gene profile revealed several useful biomarkers in TNBC, which may serve as a viable putative therapeutic target in TNBCs and suggest the possible use of TQ in the chemoprevention/treatment of TNBC.

## Figures and Tables

**Figure 1 nutrients-14-04787-f001:**
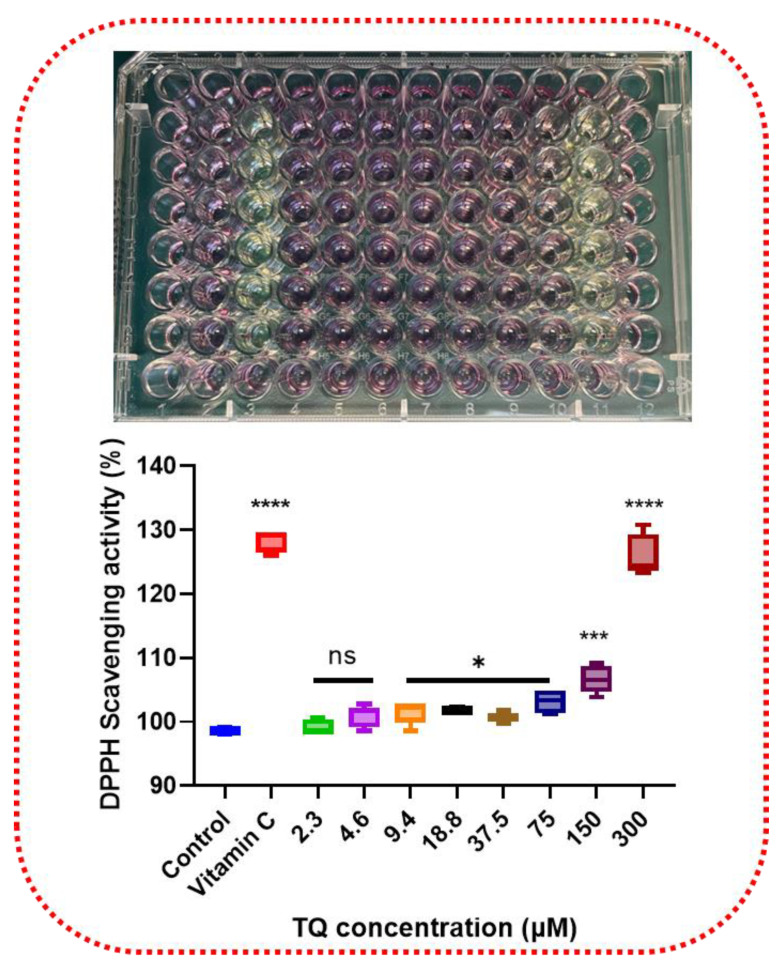
DPPH scavenging activity of TQ. The antioxidative capacities of TQ were determined by their capabilities to scavenge DPPH. DPPH was dissolved in DMSO and diluted in Methanol. TQ (0–300 μM) was reacted with DPPH. TQ scavenged DPPH in a concentration-dependent way in 30 min of reaction time. TQ was able to convert the stable radical DPPH to yellow diphenyl picrylhydrazine. Vitamin C was used as a positive control. Asterisks indicate a significant increment in DPPH scavenging activity compared with controls. Each bar indicates the mean ± SEM of three replications using one-way ANOVA followed by the Bonferroni test. statistical significance at * *p* ≤ 0.05, *** *p* < 0.001, **** *p* < 0.0001, ns non-significant. DPPH, 2,2-diphenyl-1-picrylhydrazyl; DMSO, dimethyl sulfoxide.

**Figure 2 nutrients-14-04787-f002:**
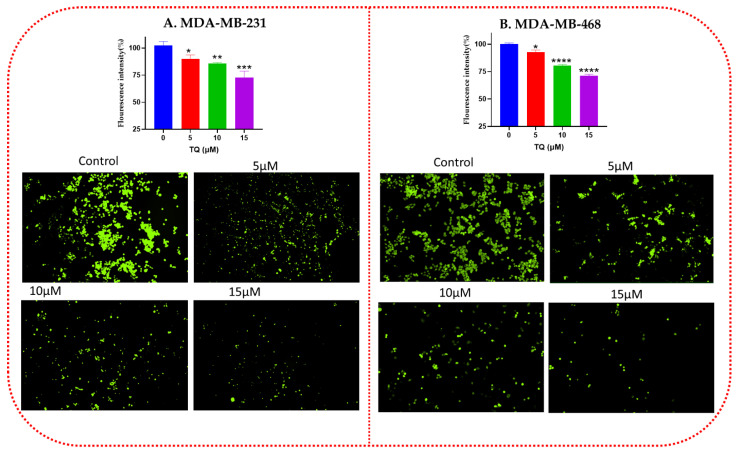
The intracellular level of ROS in TQ-treated MDA-MB-231 (**A**) and MDA-MB-468 (**B**) TNBC cells. The effects of TQ on the intracellular ROS level of TNBC cells were examined by DCFH-DA assay. TNBC cells were treated with different concentrations of TQ (5–15 µM) for 24 h, followed by 45 min incubation period with 10 µM DCFDA. TQ significantly reduced the ROS level compared to control groups in a concentration-dependent manner. A Synergy HTX Multi-Mode microplate reader (BioTek Instruments, Inc., Winooski, VT, USA) was used to measure ROS fluorescence. The green fluorescence of the reacted DCFH-DA, which indicates the ROS level, was observed with fluorescence microscopy using Nikon inverted microscope eclipse (NY 11747-3064, USA). Three independent experiments/3 replicates each was conducted to generate the data and presented as mean ± SEM. Asterisks indicate a significant reduction in ROS level compared with control cells only (* *p* < 0.05, ** *p* < 0.01, *** *p* < 0.001, **** *p* < 0.0001). ROS, reactive oxygen species; DCFH-DA, 2,7-dichlorodihydrofluorescein diacetate; TQ, Thymoquinone.

**Figure 3 nutrients-14-04787-f003:**
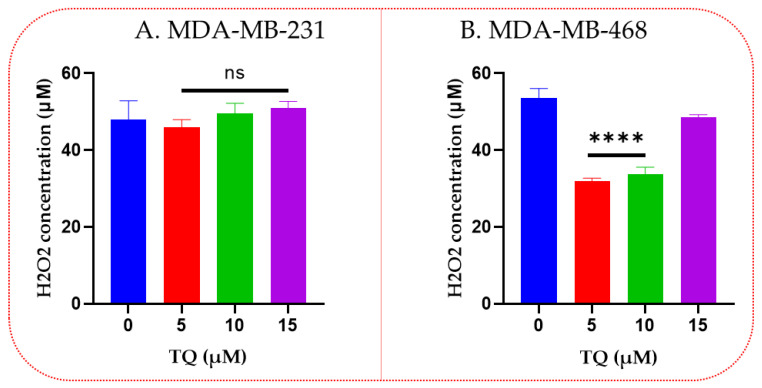
The concentration of H_2_O_2_ following TQ treatment in TNBC cells. The levels of H_2_O_2_ for MDA-MB-231 (**A**) and MDA-MB-468 (**B**) were demonstrated following treatment with TQ for 24 h at varying concentrations. The data were evaluated by one-way ANOVA. The experiment was repeated three times with *n* = 3, and its significance was indicated by the *p*-value **** < 0.0001, ns–not significance.

**Figure 4 nutrients-14-04787-f004:**
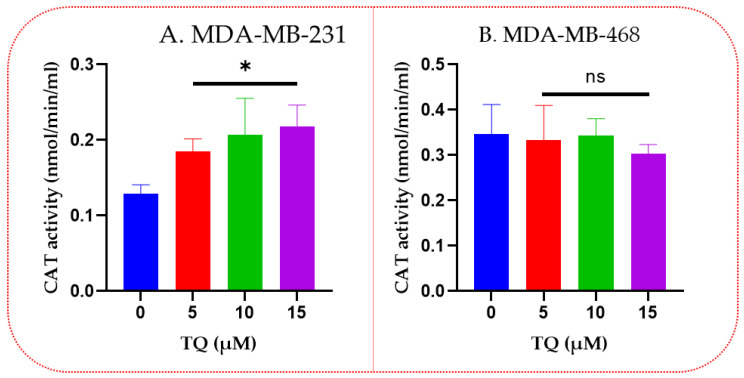
TQ impact on catalase enzyme activity in TNBC cells. 1.5 × 10^6^ cells/well/6-well plate was incubated overnight, then treated for 24 h with TQ at various concentrations (0–15 µM). Catalase activity was displayed on MDA-MB-231 TNBC cells (**A**) and MDA-MB-468 TNBC cells (**B**). Asterisks show that catalase activity has significantly increased when compared to controls. Each bar indicates the mean ± SEM of three replications using a one-way ANOVA. * *p* < 0.05 is statistical significance, and ns: nonsignificant.

**Figure 5 nutrients-14-04787-f005:**
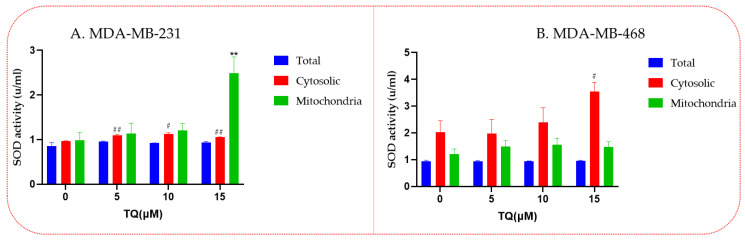
TQ effect on SOD enzyme activity in TNBC cells. In this assay, 1.5 × 10^6^ cells/well/6-well plates were incubated overnight before treatment. The next day, cells were treated for 24 h with different concentrations of TQ. SOD activity was displayed on MDA-MB-231 cells (**A**) and MDA-MB-468 cells (**B**). Asterisks show that SOD activity has significantly increased when compared to controls. Each bar indicates the mean ± SEM of three replications using one-way ANOVA. The increased levels were considered statistically significant at ** *p* < 0.01, ^#^
*p* < 0.05, ^##^
*p* < 0.01, and ns nonsignificant when comparing control vs. treated, (*) refers to the significant increase in SOD activity in the mitochondria, whereas # refers to the significant increase in SOD activity in the cytosol.

**Figure 6 nutrients-14-04787-f006:**
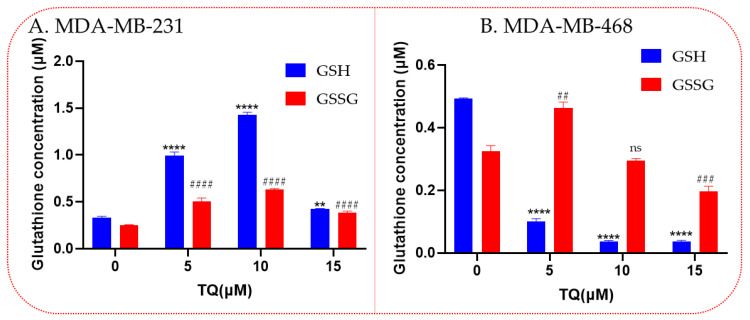
Glutathione activity in TQ-treated TNBC cells. For this study, 1.5 × 10^6^ cells/well/6-well plate was incubated at night, then treated for 24 h with TQ at various concentrations. GSH activity was displayed for MDA-MB-231 cells (**A**) and MDA-MB-468 cells (**B**). Asterisks show that Glutathione activity has significantly changed when compared to controls. Each bar indicates the mean ± SEM of three replications using one-way ANOVA. The effect is statistically significance at ** *p* < 0.01, **** *p* < 0.0001, ^##^
*p* < 0.01, ^###^
*p* < 0.001, ^####^
*p* < 0.0001 and ns; non-significant. Compared with the control, (*) refers to the significant change in GSH, meanwhile (#) refers to the significant change in GSSG.

**Figure 7 nutrients-14-04787-f007:**
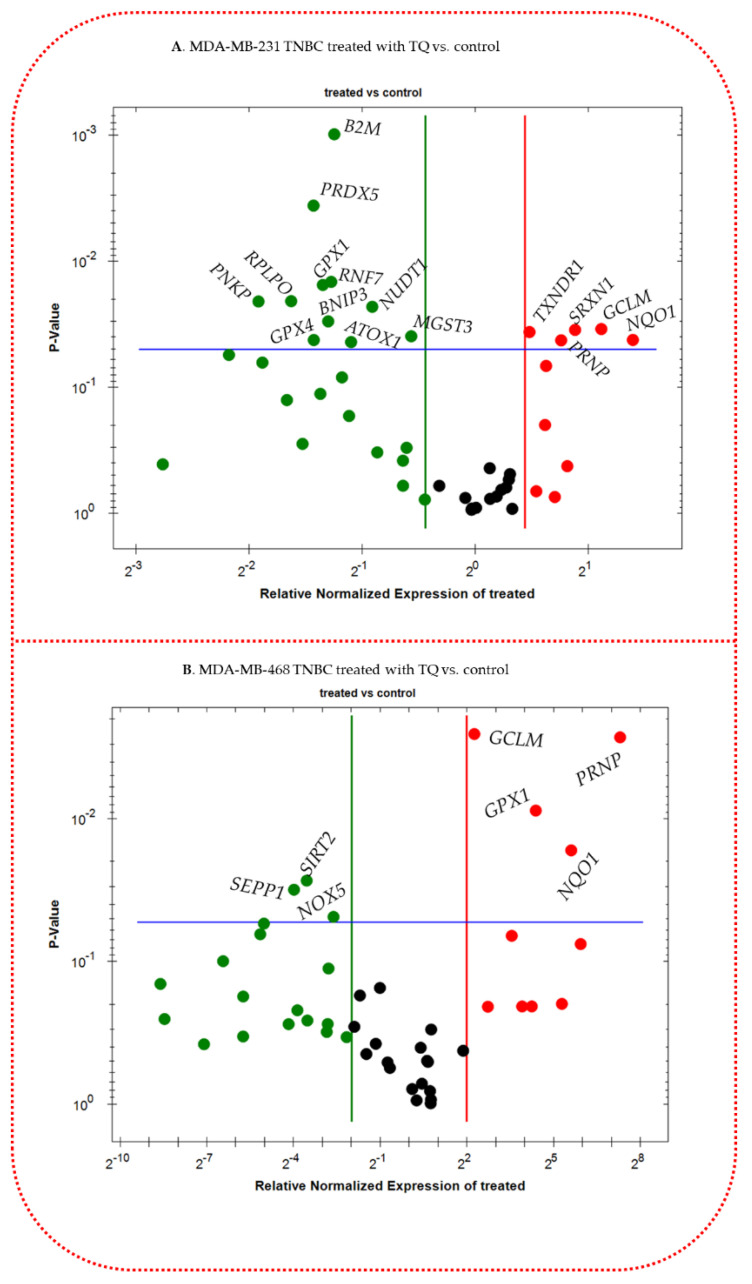
TQ altered oxidative stress−antioxidant defense-related gene expression in TNBC cells. After a 24 h treatment period with 15 µM in MDA−MB-231 cells (**A**), 5 and 11 genes were upregulated and downregulated, respectively, while in MDA−MB-468 cells (**B**) four and three genes were upregulated and downregulated, respectively. A volcano plot was used to categorize and display increased(red), repressed (green), or unaltered (black) mRNA gene expression. *GPX1*: glutathione peroxidase 1, *GPX4*: glutathione peroxidase 4 *GCLM*: glutamate−cysteine ligase, modifier subunit, *NQO1*: NAD(P)H dehydrogenase, quinone 1, *PRNP*: prion protein, *SIRT2*: sirtuin 2, *NOX5*: NADPH oxidase, EF−hand calcium binding domain 5, *TXNRD1*: thioredoxin reductase 1, *SEPP1*: selenoprotein P, plasma, 1, *PRDX5*: peroxiredoxin 5, *B2M*: beta−2−microglobulin, *RPLPO*: ribosomal protein, large, P0, *PNKP*: polynucleotide kinase 3′−phosphatase, *RNF7*: ring finger protein 7, *BNIP3*: BCL2/adenovirus E1B 19 kDa interacting protein 3, *NUDT1*: nudix (nucleoside diphosphate linked moiety X)−type motif 1, *ATOX1*: antioxidant protein 1 homolog (yeast), *MGST3*: microsomal glutathione S−transferase 3, *SRXN1*: sulfiredoxin 1.

**Figure 8 nutrients-14-04787-f008:**
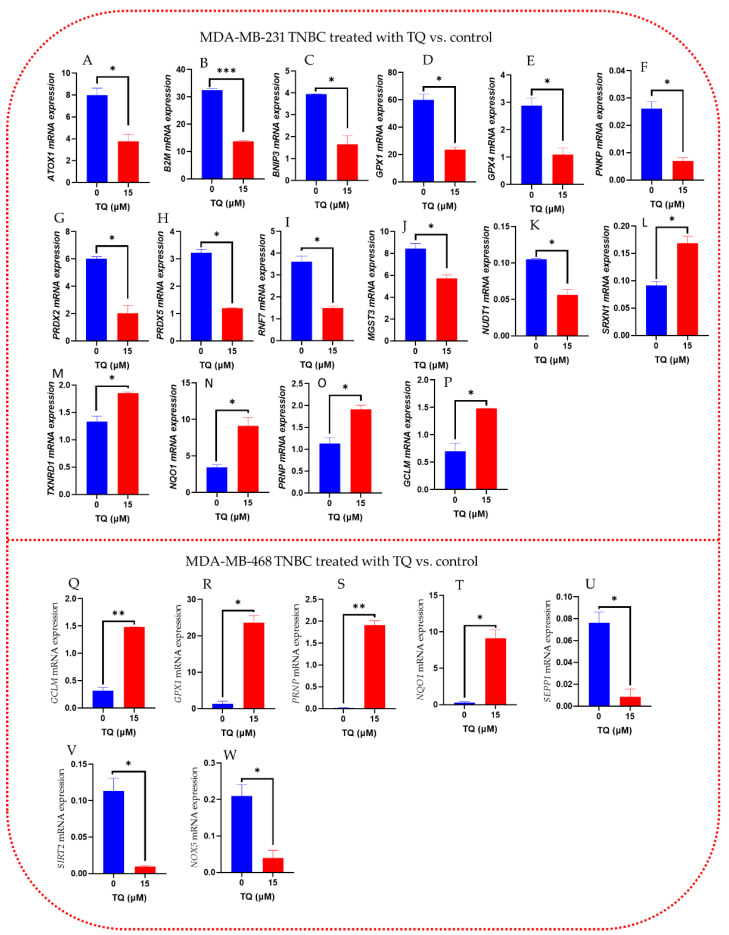
PCR analysis of the effect of TQ on the expression of genes involved in oxidative stress and antioxidant defense in MDA-MB-231 (**A**–**P**) or MDA-MB-468 (**Q**–**W**) TNBC cell lines, following 15 µM of TQ treatment in the MDA-MB-231 TNBC cell lines and the MDA-MB-468 TNBC cells. DMSO was used to treat the control cells at 0.1%. The difference in mRNA expression between the treatments and the control was used to calculate gene expression. The findings represent the means of a minimum of four biological research (*n* = 4). Using the Student’s t-test, the statistical significance of differences between the control and TQ treatments was established. The statistically significant differences were indicated by * *p* < 0.05, ** *p* < 0.01, *** *p* < 0.001.

**Figure 9 nutrients-14-04787-f009:**
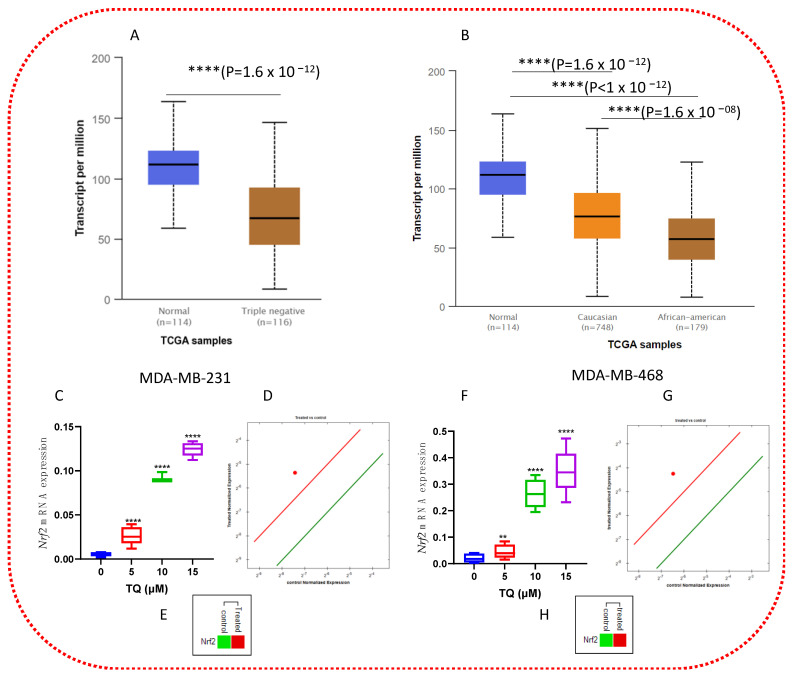
TQ’s effects on *Nrf*2 mRNA expression. UALCAN database showed the expression of Nrf2 in BRCA based on normal breast vs. TNBC (**A**) and the expression of Nrf2 in BRCA based on the patient’s race (**B**). (**C**,**F**) The box and violin graph depicted the relative *Nrf*2 mRNA expression after 5, 10, and 15 µM TQ treatment versus control after 24 h incubation. The red dot represents *Nrf*2 upregulation at 15 µM of TQ (**D**,**G**). The green color represents *Nrf*2 downregulation, while the red color represents *Nrf*2 upregulation at 15 µM of TQ (**E**,**H**). The CFX96 software was used to analyze the expression. The error bars represent the mean and the standard error of the mean. A non-parametric student t-test was used to identify significant differences (*n* = 3 per treatment in four independent experiments: ** *p* < 0.01, **** *p* < 0.0001 (comparing DMSO- and TQ-treated groups).

**Figure 10 nutrients-14-04787-f010:**
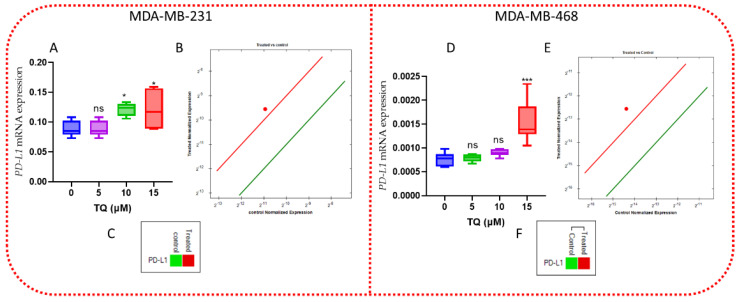
TQ’s effects on *PD*−*L*1 mRNA expression. (**A**,**D**) The box and violin graph depicted the relative *PD*−*L*1 mRNA expression after 15 µM TQ treatment versus control after 24 h incubation. The red dot represents *PD*−*L*1 upregulation at 15 µM TQ (**B**,**E**). The green color represents *PD*−*L*1 downregulation, while the red represents *PD*−*L*1 upregulation 15 µM TQ (**C**,**F**). The CFX96 software was used to analyze the expression. The error bars represent the mean and the standard error of the mean. A non-parametric student t−test was used to identify significant differences (*n* = 3 per treatment in four independent experiments: * *p* < 0.05, *** *p* < 0.001, ns-nonsignificant comparing DMSO−and TQ-treated groups).

**Figure 11 nutrients-14-04787-f011:**
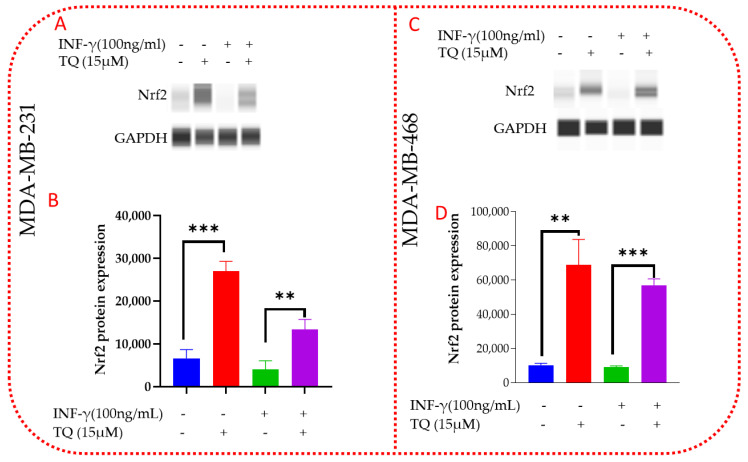
TQ modulatory effect on Nrf2 protein expression in MDA-MB-231 and MDA-MB-468 TNBC cells. Following the overnight incubation period, cells were equally treated with 15 µM of TQ for 24 h, whereas control cells were exposed only to 0.1% DMSO after overnight incubation. The Western analysis used data to evaluate the protein expression of Nrf2 in MDA-MB-231 (**A**,**B**) and MDA-MB-468 (**C**,**D**). Figures (**A**,**C**) present the cropped blots from Compass software corresponding to the protein expression of four samples treated with DMSO, TQ, INF-γ, and TQ  +  INF-γ. Additionally, (**B**,**D**) show a bar graph depicting the total Nrf2 protein expression for the same treatments. Data generated from three independent experiments were evaluated for statistical significance effects using a student t-test. ** *p* < 0.01, *** *p* < 0.001.

**Figure 12 nutrients-14-04787-f012:**
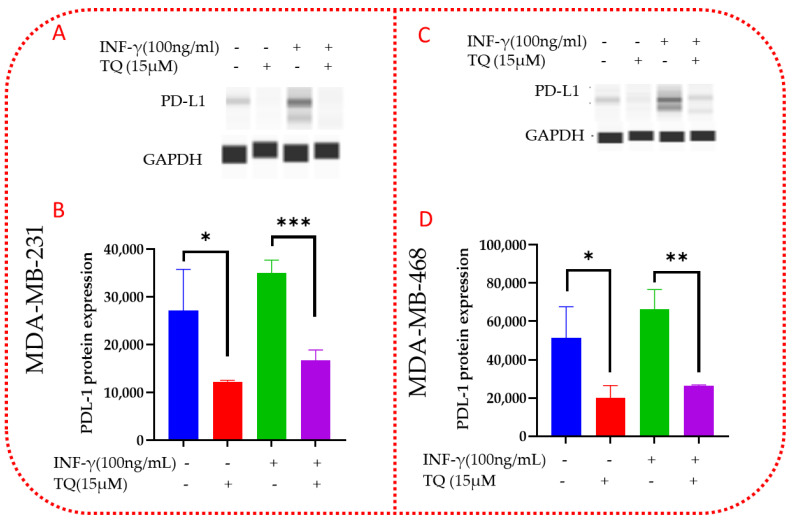
TQ modulatory effect on PD-L1 protein expression in MDA-MB-231 and MDA-MB-468 TNBC cells after 24 h treatment. Western analysis was used to evaluate the expression of the PD-L1 protein, MDA-MB-231 (**A**,**B**), and MDA-MB-468 (**C**,**D**). (**A**,**C**) shows blot view cropped from Compass software corresponding to the protein expression after exposure to the treatments. (**B**,**D**) show a bar graph depicting total PD-L1 expression after the following treatments: control, TQ, INF-γ, and TQ  +  INF-γ, respectively. TQ and INF-γ and INF-γ vs. TQ  +  INF-γ were evaluated for statistical significance using one-way ANOVA and Dunnett’s multiple comparison tests. * *p* < 0.05, ** *p* < 0.01, *** *p* < 0.001.

**Figure 13 nutrients-14-04787-f013:**
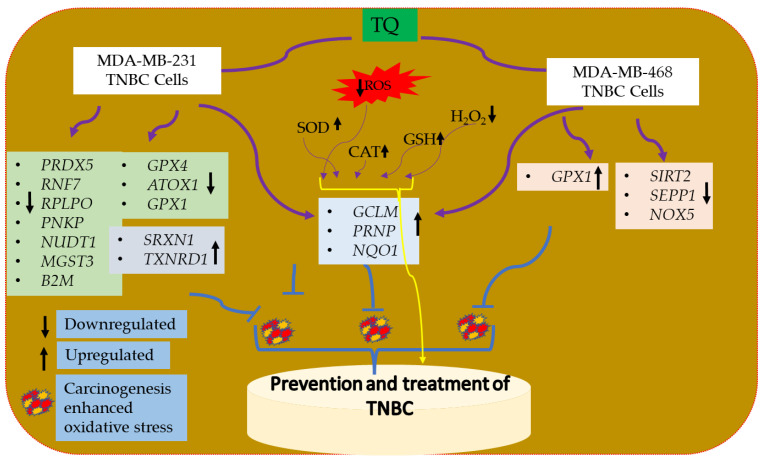
The effect of TQ on oxidative stress-antioxidant defense system gene expression in MDA-MB-231 and MDA-MB-468 TNBC cells. TQ downregulates and upregulates important cytoprotective genes. The consequence of these effects would be useful in preventing and treating TNBC. The black arrow indicates that genes are either upregulated or downregulated in TNBC cells. Altogether, it may be suggested to utilize TQ as an agent for the prevention and treatment of TNBC.

**Table 1 nutrients-14-04787-t001:** A comparative illustration of TQ effects on mRNA oxidative stress-antioxidant defense gene expression in MDA-MB-231 and MDA-MB-468 TNBC cells following a 24 h exposure period.

A. MDA-MB-231 TNBC Treated with TQ vs. Control			B. MDA-MB-468 TNBC Treated with TQ vs. Control		
GENE	Fold Change	*p*-Value	GENE	Fold Change	*p*-Value
*B2M*	−2.37	0.000986	*SIRT2*	−11.68	0.027194
*PRDX5*	−2.69	0.003623	*SEPP1*	−15.84	0.031466
*RNF7*	−2.42	0.014579	*NOX5*	−6.14	0.048628
*GPX1*	−2.54	0.015438	*PRNP*	+157.65	0.002687
*RPLPO*	−3.09	0.020751	*GCLM*	+4.78	0.002548
*PNKP*	−3.77	0.020877	*GPX1*	+20.87	0.008767
*NUDT1*	−1.88	0.022986	*NQO1*	+48.87	0.016668
*BNIP3*	−2.46	0.030008			
*MGST3*	−1.48	0.039413			
*GPX4*	−2.69	0.042205			
*ATOX1*	−2.14	0.043778			
*GCLM*	+2.17	0.034524			
*SRXN1*	+1.85	0.035065			
*TXNRD1*	+1.40	0.036500			
*NQO1*	+2.63	0.042095			
*PRNP*	+1.70	0.042419			

## Data Availability

All data generated or analyzed during this study are included in this published article.

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
