# Peer review of "Anticancer Effects of Thymoquinone through the Antioxidant Activity, Upregulation of Nrf2, and Downregulation of PD-L1 in Triple-Negative Breast Cancer Cells"

_nutrients, 2022, doi:10.3390/nu14224787_

Round 1
Reviewer 1 Report
The authors aimed to investigate the effect of thymoquinone on oxidative stress markers and antioxidant defenses in MDA-MB-231 and MDA-MB-468 TNBC cells, pointing to its effect on Nrf2 and PD-L1. While a significant amount of work has been accomplished, there is a major concern with the novelty in this manuscript since all the reported effect of TQ are well-demonstrated in different disease models and in vivo and in vitro investigations.
The abstract should be shortened since it contains a long background.
The introduction should be more focused.
Several studies have shown the effect of TQ on the viability, proliferation and migration of MDA-MB triple negative breast cancer cells (doi: 10.2174/1871520620666200807221047 ---doi: 10.5306/wjco.v12.i5.342).
The authors assayed the DPPH radicals scavenging activity which is a very well-known activity of TQ. So there was no need for this assay to support the radical-scavenging activity of TQ.
The authors then determined the effect of TQ on ROS, hydrogen peroxide, glutathione, SOD activity, and CAT activity in the cells. All of these are demonstrated effects of TQ both in vitro and in vivo as shown by several investigators.
The manuscript needs a major re-writing and to be more focused on the outcomes.
Figure 11 and 12 showing the changes in Nrf2 and PD-L1, respectively, are of very poor quality.
Author Response
Dear Editor:
We are pleased to resubmit the revised version of the Manuscript ID: nutrients-1950213 "Anticancer effects of thymoquinone through the antioxidant activity, upregulation of Nrf2, and downregulation of PD-L1 in triple-negative breast cancer cells". Getinet M. Adinew, Samia Messeha, Equar Taka, Ramesh Badisa, Karam F A Soliman *. We appreciate the reviewers' constructive criticisms, questions, and comments, and we have addressed each of their concerns as outlined below.
Reviewer 1
- The abstract should be shortened since it contains a long background.
- Response: Modified as suggested
- The introduction should be more focused.
- Response: Modified as suggested
- Several studies have shown the effect of TQ on the viability, proliferation, and migration of MDA-MB triple-negative breast cancer cells (doi: 10.2174/1871520620666200807221047 ---doi: 5306/wjco. v12.i5.342).
- Response: Included in the article as a reference "reference number 66 and 67 "to support our findings: lines 157 and 159, page 4.
- The authors assayed the DPPH radicals scavenging activity which is a very well-known activity of TQ. So, there was no need for this assay to support the radical-scavenging activity of TQ.
- Response: various studies showed that TQ had shown pro-oxidant activities; we conducted this to confirm our current study for our subsequent antioxidant biological studies.
- The authors then determined the effect of TQ on ROS, hydrogen peroxide, glutathione, SOD activity, and CAT activity in the cells. All of these are demonstrated effects of TQ both in vitro and in vivo, as shown by several investigators.
- Response: We agree on these previously activities, but we found mixed and different responses in these two genetically distinct cell lines, which helps researchers to explore more about these different cell lines of TNBC
- The manuscript needs a major re-writing and is more focused on the outcomes.
- Response: all texts are modified as suggested.
- Figures 11 and 12 showing the changes in Nrf2 and PD-L1, respectively, are of very poor quality.
- Response: improved by increasing the resolutions of figures.

Reviewer 2 Report
Manuscript Number: nutrients-1950213
Manuscript Title: Thymoquinone anticancer effects through the antioxidant activity, upregulation of Nrf2, and downregulation of PD-L1 in 3 triple-negative breast cancer cells.
Name of the Journal: Nutrients
General comments:
The Authors submitted the Manuscript entitled "Thymoquinone anticancer effects through the antioxidant activity, upregulation of Nrf2, and downregulation of PD-L1 in triple-negative breast cancer cells" submitted for possible publication in Nutrients, MDPI.
The authors have investigated the anticancer effects of Thymoquinone for the establishment of its role in breast cancer. The authors well-established the mechanism of the anticancer effect of thymoquinone through the antioxidant activity, upregulation of Nuclear factor-erythroid 2-related factor (Nrf2), and downregulation of Programmed death-ligand (PD-L1) in triple-negative breast cancer cells.
The work carried out by the authors is an excellent piece of work and affords substantial outcomes. Overall, the manuscript is well-written and all sections in the manuscript are well-placed and arranged. However, this manuscript has some minor errors, which need to be eliminated before it is fit for publication. Here are some comments for the author’s consideration.
Some specific comments are;
1. Title: The title needs to be modified to “Anticancer effects of thymoquinone through the antioxidant activity, upregulation of Nrf2, and downregulation of PD-L1 in triple-negative breast cancer cells”.
2.There are some syntax errors and typographical mistakes in some places in the manuscript. Hence, these errors and mistakes shall be corrected before publication.
Abstract:
3. There is no methodology in the abstract section, hence incorporate a methodology in brief in the abstract section (In line 19, after and MDA-MB-468 TNBC cells.).
4. In line 21, delete, Catalase (CAT).
5. Write a graphical abstract if possible, for a better presentation of the abstract.
Introduction:
6. Give the full form of all abbreviations at the first appearance in the text and the used abbreviation at the next appearance in the manuscript. For example, HER2 in line 49, TNBC in line 54, ROS in line 60, GSH in line 76, etc.
Materials and methods:
2.2 Cell culture
7. Line-186, line, & line 190, mL instead of ml.
8. Similarly, use μL instead of μl.
9. Use of the uppercase letter L at all appropriate places of units, like ml and μl.
Results:
10. In figure-1, at X-axis, write TQ concentration (µM) instead of TQ (µM), above line 332. Similarly, in all other figures, if possible.
11. Line 337, write Vitamin C instead of Vit. C.
12. It is also suggested that dpi of figures shall be increased for better visibility, again if possible.
Discussion and Summary
13. Discussion and Summary are very well written.
References:
14. Some references are very old. Better to replace those old references with new ones, preferably from the last five years or at least within the last 10 years.
Overall review Comments: Authors should address the above comments the and manuscript needs minor revision to publish in your esteemed Journal

Author Response
Reviewer 2
- Title: The title needs to be modified to "Anticancer effects of thymoquinone through the antioxidant activity, upregulation of Nrf2, and downregulation of PD-L1 in triple-negative breast cancer cells".
- Response: Modified as suggested, included on page 1, lines 2-4
- There are some syntax errors and typographical mistakes in some places in the manuscript. Hence, these errors and mistakes shall be corrected before publication.
- Response: Checked in the whole text
- There is no methodology in the abstract section; hence incorporate a method in brief in the abstract section (In line 19, after and MDA-MB-468 TNBC cells.).
- Response: included as suggested as "The DPPH assay, ROS assay, various antioxidant activities, and the expression of the genes for Nrf2 and PD-L1 are all included in these in vitro research", page 1, line 22-23.
- In line 21, delete, Catalase (CAT).
- Response: deleted as suggested.
- Write a graphical abstract, if possible, for a better presentation of the abstract.
- Response: included as suggested.
- Give the full form of all abbreviations at the first appearance in the text and the used abbreviation at the next appearance in the manuscript. For example, HER2 in line 49, TNBC in line 54, ROSin line 60, GSH in line 76, etc.
- Response: included as suggested Page 2, Line 52 for TNBC, line 55 for HER2, line 66 for ROS, and line 74 for GSH.
- Line-186, line, & line 190, mL instead of ml.
- Response: included as suggested Page 4, lines 200 and 202 as" Both cell lines were grown in 75-mL tissue culture (TC) flasks as monolayers at 37 °C in a humidified 5% CO2 incubator, occasionally subculturing with trypsin/EDTA. 4 mM L-glutamine".
- Similarly, use μL instead of μl.
- Response: corrected as suggested on page 5, lines 210-214
- Use the uppercase letter L at all appropriate places of units, like ml and μl.
- Response: corrected as suggested
- In figure 1, at the X-axis, write TQ concentration (µM) instead of TQ (µM), above line 332. Similarly, in all other figures, if possible.
- Response: corrected as suggested, page 8, line 350.
- Line 337, write Vitamin C instead of Vit. C.
- Response: corrected as suggested, page 8, lines 348 and 355.
- It is also suggested that the dpi of figures shall be increased for better visibility, again, if possible.
- Response: corrected as suggested, Figure 1-12
- Discussion and Summary are very well written.
- Some references are very old. Better to replace those old references with new ones, preferably from the last five years or at least within the last 10 years.
- Response: corrected as suggested all references are = > 2012
Overall review Comments: Authors should address the above comments, and the manuscript needs minor revision to publish in your esteemed Journal

Reviewer 3 Report
The manuscript entitled “Thymoquinone anticancer effects through the antioxidant activity, upregulation of Nrf2, and downregulation of PD-L1 in triple-negative breast cancer cells” is a valuable study in the field of antioxidant nutrients and bioactive compounds in the prevention of chronic diseases.
As the reviewer is obliged to pay attention to the following elements:
line 21: phrase "Catalase (CAT)." is misunderstandable;
line 48: it will be better to write "lower" than "overall lower";
general: please use appropriate degree sign;
general: unify figure references (there is in manuscript Fig 2A, Fig. 2B, figure 3A and Figure 4A);
Author Response
Reviewer 3
The manuscript entitled "Thymoquinone anticancer effects through the antioxidant activity, upregulation of Nrf2, and downregulation of PD-L1 in triple-negative breast cancer cells" is a valuable study in the field of antioxidant nutrients and bioactive compounds in the prevention of chronic diseases.
As the reviewer is obliged to pay attention to the following elements:
- Line 21: the phrase "Catalase (CAT)." is misunderstandable;
- Response: deleted, line 26, page 1
- line 48: it will be better to write "lower" than "overall lower";
- Response: corrected as suggested, included on page 2, line 54 as "…and has a worse overall survival rate."
- general: please use the appropriate degree sign;
- Response: corrected as suggested on page 6, lines 291-306.
- general: unify figure references (there are in manuscript Fig 2A, Fig. 2B, figure 3A and Figure 4A);
- Response: corrected, all figure references on the discussion part are deleted.

Round 2
Reviewer 1 Report
The authors did not address the comments adequatly.
There is still a major concern regarding the novelty of the study because these findings have been previously reported as mentioned in my previous report.
The manuscript needs major rewriting, in particular the abstract and introduction sections. The abstract, introduction, and discussion sections are very long.
The authors stated that they improved the quality of Figures 11 and 12, but the quality is very poor and unsuitable for publication. The authors divided the blots into separate bands which is very questionable. The cropped bands "non-separated" should be presented in the manuscript and the full blots should be provided as supplementary. Actually, it is very hard to see the bands and hence the quantification could not be judged.
The authors replied to comment number 3 " Several studies have shown the effect of TQ on the viability, proliferation, and migration of MDA-MB triple-negative breast cancer cells (doi: 10.2174/1871520620666200807221047 ---doi: 10.5306/wjco. v12.i5.342)."by citing these references in the manuscript, but this was not the question. This comment shows that the findings in this study are not novel and most of them have already been previously reported. Still can not see any novel addition in this study.
The response to comment number 4 "The authors assayed the DPPH radicals scavenging activity which is a very well-known activity of TQ. So, there was no need for this assay to support the radical-scavenging activity of TQ." is not acceptable because the antioxidant and prooxidant of TQ and any other compound depend on its microenvironment. Did the authors expect that TQ might exert a prooxidant effect when incubated in an environment containing DPPH radicals?
Author Response
Dear Editor:
- We are pleased to resubmit the revised version of the Manuscript ID: nutrients-1950213 "Anticancer effects of thymoquinone through the antioxidant activity, upregulation of Nrf2, and downregulation of PD-L1 in triple-negative breast cancer cells". Getinet M. Adinew, Samia Messeha, Equar Taka, Ramesh Badisa, Karam F A Soliman *. We appreciate the constructive criticisms, questions, and comments of the reviewers, and we have addressed each of their concerns as outlined below. Also, please note that the manuscript was totally revised and edited
- Note: -We included separated supplement material containing the whole blot of western analysis for protein expression of Nrf2 and PD-L1.
Reviewer 1-second round
- There is still a major concern regarding the novelty of the study because these findings have been previously reported, as mentioned in my previous report.
- Response: We included a paragraph that showed the novelty of the study Pag2 Line 55-76 as "…. cancer cells may produce more ROS because they lack the genes that normally produce antioxidant defenses…. Continuous production of ROS by the tumor cells causes increased mutation rates and accelerated tumor progression, activation of signaling pathways that promote growth, adaptation to oxidative stress leading to increased therapy resistance, and increased blood supply to the tumor cells. Increased risk of metastasis…. promising treatment for TNBC", and sentences page 3, line 97-102 as" Our study is interesting because, to the best of our knowledge, we are the first to demonstrate the expression profiles of the genes associated with the antioxidant defense system against oxidative stress as well as the effects of TQ on Nrf2 and PD-L1 in MDA-MB-231 and MDA-MB-468 TNBC cells.".
- The manuscript needs major rewriting, in particular the abstract and introduction sections. The abstract, introduction and discussion sections are very long.
- Response: Modified as suggested.
- the abstract is shortened from 405 to 246 words, page 1, lines 11-30
- the introduction is shortened from 1572 to 850-words, lines 35-102
- Response: Modified as suggested.
- the discussion part was shortened from 4123 to 2086 words, lines 547-715.
- The authors stated that they improved the quality of Figures 11 and 12, but the quality is very poor and unsuitable for publication. The authors divided the blots into separate bands, which is very questionable. The cropped bands "non-separated" should be presented in the manuscript, and the full blots should be provided as supplementary. Actually, it is very hard to see the bands, and hence the quantification could not be judged.
- Response: Modified figures 11 and 12 as suggested. The full blots are provided in a separate document as a supplementary.
- The authors replied to comment number 3 " Several studies have shown the effect of TQ on the viability, proliferation, and migration of MDA-MB triple-negative breast cancer cells (doi: 10.2174/1871520620666200807221047 ---doi: 5306/wjco. v12.i5.342)." by citing these references in the manuscript, but this was not the question. This comment shows that the findings in this study are not novel, and most of them have already been previously reported. Still cannot see any novel addition in this study.
- Response: We included sentences (page 2, line 55.76) that showed the novelty of the study. Pag2-3 Line 97-102.as" Our study is interesting because, to the best of our knowledge, we are the first to demonstrate the expression profiles of the genes associated with the antioxidant defense system against oxidative stress as well as the effects of TQ on Nrf2 and PD-L1 in MDA-MB-231 and MDA-MB-468 TNBC cells.".
- The response to comment number 4 "The authors assayed the DPPH radicals scavenging activity which is a very well-known activity of TQ. So, there was no need for this assay to support the radical-scavenging activity of TQ." is not acceptable because the antioxidant and prooxidant of TQ and any other compound depend on its microenvironment. Did the authors expect that TQ might exert a prooxidant effect when incubated in an environment containing DPPH radicals?
- Response: As the reviewer stated, Thymoquinone has both antioxidant and prooxidant activities, but as our research is focused on its cytoprotective effects on TNBC cells, we first tried to see if it had an antioxidant activity with the concentrations used in our subsequent studies. We should do the DPPH assay to demonstrate that TQ's anticancer activity is caused by its antioxidant characteristics rather than by its prooxidant activities, as various research has revealed that TQ has both antioxidant and prooxidant activities. We used a variety of other metrics to confirm the antioxidant activity further after the DPPH assay was successful. As the reviewer stated, our findings of antioxidant activities were consistent with the earlier findings. Our research on antioxidants identified a gene involved in controlling genes involved in the oxidative-antioxidant defense system, which was supported by its effect on Nrf2 expression.
